# ATTENTIONR-GCN: INCORPORATING SPATIOTEMPORAL REASONING IN HETEROGENEOUS AND PARTIALLY OBSERVED GRAPHS

## ABSTRACT

Urban infrastructure networks are complex systems characterized by heterogeneous nodes and edges, partial observability, and temporal dynamics, which many graph neural networks struggle to handle. We introduce AttentionR-GCN, an extension of graph attention network based on different relational types that (i) uses attention-based message aggregation to weight node and edge signals under different relation types, (ii) uses learnable embeddings to represent missing values, and (iii) incorporates a transformer encoder to model temporal dependencies. We evaluate AttentionR-GCN on two simulated water distribution networks, predicting one-step-ahead chlorine concentrations at both monitored and unmonitored nodes under varying levels of missing sensor data. Our model outperforms different baselines, especially under high data sparsity, and demonstrates superior generalization to unmonitored nodes. Our results reveal the importance of incorporating adaptive weighting of node and edge features under different relations, learnable representations for missing values, and capturing temporal dependencies to achieve more reliable predictions in partially observed infrastructure networks.

## 1 INTRODUCTION

Urban infrastructure networks, such as water distribution systems, electrical grids, and transportation systems, form the backbone of modern cities. These networks are complex and consist of heterogeneous components that exhibit dynamic temporal-spatial behavior, often operating under limited sensing coverage. Monitoring the status of individual components within these networks is critical for ensuring system stability. In practice, soft sensors are used to estimate physical parameters at key locations throughout the network. However, due to the high costs and practical challenges, coverage is often limited to a small subset of the network, leaving large portions unmonitored and uncertain. Consequently, these networks are partially observed, with critical physical attributes missing at many locations and time steps. This sparsity, combined with the heterogeneity of node and edge types and complex temporal dependencies, presents a significant challenge for traditional machine learning models.

Graph neural networks are well suited for urban networks because they propagate information across interconnected components. In homogeneous scenarios, GAT Veličković et al. (2017) and GATv2 Brody et al. (2021) learn neighbor-specific attention weights through multi-head aggregation to create expressive, denoising message passing with strong inductive generalization to unseen nodes. In heterogeneous settings, R-GCN Schlichtkrull et al. (2018) captures multi-relational structure via relation-specific transformations with shared parameters, allowing efficient training and strong performance. Extending attention to heterogeneous graphs, r-GAT Chen et al. (2021) and RGAT Busbridge et al. (2019) directly incorporate relation information into the attention mechanism to better utilize multi-relational information.

However, these models are inadequate for multi-relational, spatio-temporal urban data. GAT and GATv2 are designed for homogeneous graphs and do not support multiple relation types; R-GCN models relations but do not incorporate continuous edge attributes into messages or attention. r-GAT and RGAT add relation conditioning, yet treat relations as categorical labels and only weakly couple relational signals with edge features, leaving rich edge attributes underused. As a result,

none of these methods fully integrate relation type, edge features, and node features within a unified attention or message pathway. Moreover, they lack explicit mechanisms to handle severe observation sparsity and to capture temporal dynamics, both of which are crucial in real-world urban networks.

To address these limitations, we introduce AttentionR-GCN, a unified architecture that combines relation-aware attention for message passing, learnable mechanisms for missing-data handling, and a Transformer-based temporal encoder. This design enables end-to-end reasoning over heterogeneous graph structures, reliable interpolation under observation sparsity, and robust modeling of temporal dependencies. We evaluate AttentionR-GCN on two realistic water-distribution simulations to forecast chlorine concentration across all nodes under varying sparsity levels. Across metrics, AttentionR-GCN consistently outperforms GAT, GATv2, R-GCN, RGAT, and their Transformer-augmented variants, especially in medium to high sparsity regimes and on unmonitored nodes. This demonstrates our model's improved robustness and generalization in challenging inference settings. Our contributions are as follows.

- **Attention-based relational aggregation**: We introduce a triplet attention mechanism that jointly considers neighboring node features, edge features, and self-node features under different relation types, enabling fine-grained, adaptive message passing over heterogeneous infrastructure graphs.

- **Learnable embeddings for missing values**: We introduce learnable parameters to represent missing inputs, enabling models to maintain predictive robustness under data sparsity.

- **Temporal modeling via transformer encoder**: We integrate a transformer module to encode multivariate time series, enabling the model to capture long-range temporal dependencies and improve forecast accuracy across time.

## 2 RELATED WORK

### 2.1 R-GCN

R-GCNs extend traditional GCNs by introducing relation-specific weight matrices to capture heterogeneous relational structures during message passing, making them suitable for graphs with multiple edge types Schlichtkrull et al. (2018). This represents an improvement over GNNs that share a single weight matrix across all edge types. R-GCNs have been successfully adapted to domains such as recommendation systems Yan et al. (2023), knowledge graph reasoning Schlichtkrull et al. (2018) and financial prediction Jeyaraman et al. (2024).

### 2.2 GAT AND GATv2

GAT and GATv2 expand convolutional GNNs by replacing fixed neighbor aggregation with learnable, nodes-edges specific attention that determines how much each neighbor contributes during message passing Veličković et al. (2017); Brody et al. (2021). GAT computes an attention logit by applying a shared learnable parameter per head to the concatenation of linearly projected features from the target node, a neighbor node, and, when available, a learned projection of the edge features between them. These logits are softmax-normalized over the target's neighborhood to produce coefficients that weight incoming messages Veličković et al. (2017). GATv2 increases scorer expressivity by inserting a nonlinearity into an attention MLP applied to a joint transformation of target, neighbor, and edge features, yielding content-dependent coefficients beyond the original linear form Brody et al. (2021). Variants of GAT and GATv2 have been deployed widely, including in transportation and spatio-temporal traffic forecasting Zhao et al. (2023); Kong et al. (2020); Chen et al. (2023) and in molecular property prediction Xu et al. (2023).

### 2.3 RGAT AND R-GAT

Relational extensions of GAT adapt neighborhood weighting to multi-relational graphs by conditioning on edge types or relation embeddings. RGAT Busbridge et al. (2019) augments GAT with relation-type parameterization, making both attention and message transformations depend on the relation. RGAT has been applied to typed-edge settings such as molecular interaction graphs, where

different bonds carry distinct semantics Busbridge et al. (2019). r-GAT Chen et al. (2021) conditions a shared attention scorer on learned relation embeddings, yielding a parameter-efficient way to model multi-relational structure. r-GAT has been used for knowledge-graph link prediction and entity classification Chen et al. (2021). Collectively, these methods show that injecting relation information into attention improves representation learning on heterogeneous graphs, especially in domains with rich relational structure.

## 2.4 GRAPH NEURAL NETWORKS (GNNs) FOR INFRASTRUCTURE MODELING

GNNs have become a central approach for spatiotemporal forecasting and state estimation in urban infrastructure, with rapid adoption in water systems and established use in transportation and energy. In water distribution networks, recent research applies GNNs to reconstruct unmonitored water-quality states and related variables under sparse sensing Salem et al. (2024). In transportation, models such as STFGNN, which fuses spatial and temporal graphs Li & Zhu (2021), Graph WaveNet, which combines adaptive learned adjacency with dilated temporal convolutions Wu et al. (2019), and GMAN, which uses multi-head spatiotemporal attention Zheng et al. (2020), illustrate how learned dependencies and temporal encoders can be effectively integrated for traffic forecasting. In energy, emerging methods pair graph encoders with Transformer-style temporal stacks for multi-site PV power forecasting, further highlighting the importance of learned, time-varying dependencies Yang et al. (2025).

Current approaches usually combine a spatial graph operator to capture flow-like interactions with a temporal encoder to model long-range dynamics, and some include edge-aware message functions. However, two gaps remain: (i) Relational heterogeneity is under-modeled, as real systems have multiple pipe, road, or feeder types and operational roles that require relation-specific message passing. (ii) Data sparsity remains a significant challenge, as limited or intermittent sensing reduces robustness, especially at unmonitored nodes. Overcoming these issues involves developing models that simultaneously encode edge types and continuous edge attributes within the message function and remain accurate under high sparsity, focusing on reliable inference at unobserved locations.

## 3 MODEL ARCHITECTURE

We propose AttentionR-GCN, which extends GATv2 by incorporating relation-aware attention inspired by R-GCN. The attention weights are conditioned on relation types as well as node and edge attributes, enabling the model to assign fine-grained importance to neighbors and capture heterogeneity across relations. To improve robustness under sparse observations, we add learnable embeddings that impute missing node features during both training and inference. We also couple the graph module with a Transformer-based encoder to model long-horizon dynamics.

### 3.1 ATTENTION-BASED AGGREGATION FOR RELATIONAL MESSAGE PASSING

GAT and GATv2 operate on homogeneous graphs and do not model multiple relation types. R-GCN handles multi-relational structures by computing relation-specific messages, but fails to utilize edge features. r-GAT primarily conditions its attention scorer on relation embeddings, but it shares parameters across relations and often uses a single softmax for all. Additionally, it only encodes relation type and not per-edge features. RGAT employs relation-specific parameters but treats relations as categorical edge types and does not propagate edge attributes through the message-passing process.

We therefore propose a relation-aware extension of GATv2 that conditions attention on relation type and directly incorporates nodes and edge attributes. For each relation $r$, attention is computed over the triplet $[x_u \, \| \, x_v \, \| \, e_{uv}]$ via a GATv2 scorer, while messages follow R-GCN by projecting the neighbor feature with a relation-specific $W_r$ and scaling by the learned coefficients. This design jointly encodes self node-edge-neighbor node under different relations and enables head-wise, fine-grained selection of informative signals in heterogeneous graphs. The specifics are as follows:

1. **Relation-specific message projection.**

$$m_{uv}^r \,=\, x_u W_r,$$

where $x_u \in \mathbb{R}^{d_{\text{in}}}$ is the neighbor node feature and $W_r \in \mathbb{R}^{d_{\text{in}} \times d_{\text{out}}}$ is a relation-specific linear projector shared across heads.

2. **GATv2-style attention over self node–neighbor node –edge features.**

$$c_{uv} = [\, x_u \,\|\, x_v \,\|\, e_{uv} \,] \in \mathbb{R}^{D_{\text{cat}}}$$

$$z_{uv}^r[he] = \text{LeakyReLU}\big(M_r^{[he]}\, c_{uv}\big),$$

$$S_{uv}^r[he] = (w_r^{[he]})^\top z_{uv}^r[he] + b_r^{[he]}.$$

where $u$ and $v$ are the neighbor and self nodes ; $r$ is the relation type; $he$ is the attention head index; $x_u, x_v \in \mathbb{R}^{d_{\text{in}}}$ are the neighbor and self nodes features; $e_{uv} \in \mathbb{R}^{d_e}$ is the edge feature; $c_{uv} \in \mathbb{R}^{D_{\text{cat}}}$ is the concatenated attention input; $d_{\text{in}}$ is the node feature dimension and $d_e$ is the edge feature dimension; $D_{\text{cat}} = 2d_{\text{in}} + d_e$ is the concatenation dimension; $M_r^{[he]}$ is the per-relation and head linear map; $d_h$ is the hidden dimension of the attention scorer; $z_{uv}^r[he]$ is the LeakyReLU activation output; $w_r^{[he]}$ and $b_r^{[he]}$ are the per-relation and head weight and bias that map $z$ to a scalar; and $S_{uv}^r[he]$ is the unnormalized attention logit for $(u, v)$, relation $r$, head $he$.

3. **Per-head normalization over neighbors of the self node.**

For each self node $v$ and head $he$,

$$a_{uv}^r[he] = \frac{\exp\big(S_{uv}^r[he]\big)}{\sum_{k \in \mathcal{N}(v)} \exp\big(S_{kv}^r[he]\big)},$$

where $a_{uv}^r[he] \in (0, 1)$ is the normalized attention coefficients that weighs neighbor $u$ relative to all neighbors of $v$.

4. **Aggregation and update at the self node.**

Weighted neighbor messages are concatenated across heads and summed as:

$$\widetilde{h}_v = \sum_{u \in \mathcal{N}(v)} \big\|_{he=1}^{H} (a_{uv}^r[he]\, m_{uv}^r) \in \mathbb{R}^{H d_{\text{out}}}$$

where $H$ is the number of heads, $\|_{he=1}^{H}$ denotes head-wise concatenation

### 3.2 LEARNABLE PARAMETERS FOR MISSING VALUES

Real-world infrastructure networks experience data sparsity due to incomplete sensor coverage. Existing approaches replace missing features with zeros or predefined constantsSalem et al. (2024); Li et al. (2024), which can introduce biases in learned representations as these constants are treated as meaningful signals. To address this issue, we propose incorporating learnable embeddings for missing node features. These embeddings are trained simultaneously with the model parameters, which allows the network to infer contextually meaningful representations for unknown data points.

### 3.3 TRANSFORMER-BASED TEMPORAL ENCODING

Standard GNN architectures do not directly integrate temporal information, which results in suboptimal performance in tasks that involve time-dependent data. We therefore incorporate a transformer-based encoder that processes sequentially structured node and edge features before relational aggregation. This transformer encoder consists of positional encodings, multi-head self-attention, and fully connected feed-forward layers.

### 3.4 COMPLETE ATTENTIONR-GCN MODEL ARCHITECTURE

The complete AttentionR-GCN model works as follows(Figure 1): first, input node and edge features, along with learnable embeddings for missing values, are processed through the transformer encoder to obtain contextualized representations. These temporal embeddings are then passed into a stack of attention-based relational graph convolutional layers. Each layer implements relational message passing using the attention-based aggregation method, followed by batch normalization,

ReLU activation, and skip connections to facilitate gradient flow. Finally, another batch normalization is added, and a linear projection maps the resulting node representations to the prediction targets. Our experiments utilize a two-layer transformer encoder with 128 hidden dimensions and 2 heads, a dropout rate of 0.2, and two attention-based R-GCN layers, each using 2 attention heads and 128 hidden dimensions.

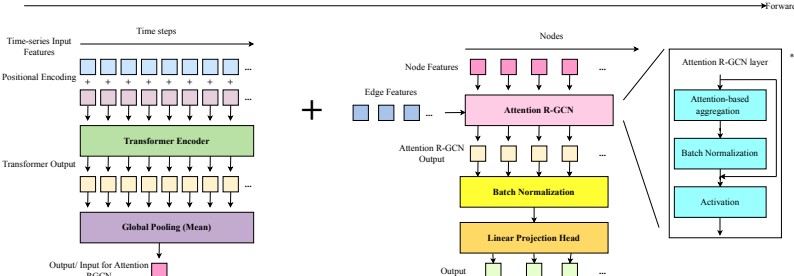

Figure 1: AttentionR-GCN model structure

### 3.5 BASELINE MODELS

We compare our model with baselines including R-GCN, GAT, and GATv2, RGAT and r-GAT, and their Transformer-augmented versions to isolate the effect of the proposed relation-aware attention. For consistency, all graph encoders use two building blocks, hidden dimension of 128, and ReLU activations. When included, the temporal module is a Transformer with an identical configuration across models.

To further assess robustness, we benchmark commonly used GNN models, including Graph Convolutional Network (GCN) Kipf (2016), Topology Adaptive Graph Convolutional Network (TAGCN) Du et al. (2017), and edge-aware model Unified Message Passing (UniMP) Shi et al. (2020), along with their transformer-augmented versions. All models used two GNN layers, a 128-dimensional hidden size, and ReLU activations.

## 4 EXPERIMENT

### 4.1 DATA

We use water distribution networks as a case study to predict next time-step chlorine concentration at whole net. Chlorine serves as a key metric for assessing water quality in water distribution networks Lipiwattanakarn et al. (2021). Our simulations are primarily based on the C-Town benchmark network, a widely used case for water distribution system studies Brahmbhatt et al. (2023); Tornyeviadzi et al. (2024); Ostfeld et al. (2012). The C-Town system comprises 5 District Metered Areas, each equipped with independent pumping stations, and sustains an average monthly total demand of 175 L/s. The network infrastructure consists of 396 nodes with 388 junctions, 1 reservoir, and 7 storage tanks, and 444 edges including 429 pipes, 11 pumps, and 4 valves.

Chlorine is introduced at the reservoir node. We simulate 2000 distinct operational scenarios by varying node demands and chlorine injection rates, following the setup from paper Salem et al. (2024). For each node, demand is generated according to Eq 1 and chlorine injection rate is generated based on Eq 2. Each simulation runs for 24 hours with a 5-minute time step. For analysis, we used the data from the last 3 hours that reach the hydraulic steady state. For each node, the simulator outputs time series water demand, hydraulic head, pressure, and chlorine concentration; for each edge, the simulator outputs time series velocity, head loss, and flow rate.

$$d(n) = \begin{cases} \dfrac{r_n}{\sum r_n} \times D, & \text{if } n \in L \subset \{1, 2, \ldots, N\}, \ |L| = U(z_{\min}, z_{\max}) \\ 0, & \text{if } n \notin L \end{cases} \tag{1}$$

where $z_{\min} = 338$ and $z_{\max} = 388$, and $d(n)$ represents the water demand at node n.

$$I(n) = \begin{cases} U(i_{\min}, i_{\max}), & \text{if } n \in m \subset \{1, 2, \ldots, N\} \\ 0, & \text{if } n \notin m \end{cases} \tag{2}$$

where $i_{\min} = 1\,\text{mg/L}$ and $i_{\max} = 4\,\text{mg/L}$, and $I(n)$ represents the injection rate at node n. In this study, N=1 and chlorine injection occurs only at the reservoir node.

We additionally evaluate on L-Town, another benchmark water distribution network widely used in prior studies Wang et al. (2025). We follow the same experimental protocol as for C-Town. L-Town contains 2 reservoirs, 782 junctions, 1 tank, 905 pipes, 1 pump, and 3 valves. Although L-Town is slightly larger than C-Town, it exhibits fewer relation types. Therefore, we primarily present results on C-Town, noting that the findings on L-Town align with those on C-Town.

## 4.2 INPUT AND OUTPUT

We frame the task as node-wise temporal prediction over a water distribution network graph. Each graph node is represented by four time-series features: water demand, hydraulic head, pressure, and chlorine concentration. Each variable spans 36 time steps, including the time 0 step, resulting in an input of size $4 \times 36$ for each node. Similarly, each edge is characterized by three temporal features: velocity, head loss, and flow rate, each spanning 36 time steps, resulting in an input of size $3 \times 36$ for each edge.

Our goal is to predict the chlorine concentration at each node after the final time step (in this case, $t = 37$), given complete temporal context up to $t = 36$. To ensure comparability, all models receive equivalent predictive context, with architectural differences determining how they leverage node and edge information. All input features and target outputs are standardized using Z-score normalization and no future information is leaked during training or evaluation. For models with a Transformer module, inputs are tensors of shape $(N, F, D)$, where $N$ is the number of nodes, $F$ the number of feature channels, and $D$ the per-feature dimensionality (temporal length). For the naive GNN baselines, we flatten the feature axis and use inputs of shape $(N, F \times D)$.

## 4.3 TRAINING AND EVALUATION SETUP

We evaluate all models on 2,000 synthetic water distribution network graphs generated from the C-Town and L-Town benchmark. The data is randomly partitioned into training (70%), validation (15%), and test (15%) splits. The test set is held out for final evaluation and is not used during model development or hyperparameter tuning. All models are trained and validated on the same splits and evaluated on the identical test set to ensure fair comparison.

Models are trained to minimize the Mean Absolute Error (MAE) between predicted and ground-truth chlorine concentrations at the final time step. Optimization is performed using the Adam optimizer with an initial learning rate of 0.001 and a ReduceLROnPlateau scheduler, which reduces the learning rate by a factor of 0.9 if validation loss does not improve for 2 consecutive epochs. Training is conducted for up to 30 epochs, with early stopping applied if validation performance does not improve for 10 consecutive epochs to mitigate overfitting. A batch size of 16 is used for training, validation, and testing.

To evaluate predictive robustness, we report three metrics: Mean Absolute Error (MAE), Mean Squared Error (MSE), and the coefficient of determination ($R^2$). MAE captures the average absolute error but treats all deviations equally. Therefore, we employed MSE that penalizes larger errors more heavily to detect whether large error occurs. However, MAE and MSE only evaluate model's predictive accuracy. We employed $R^2$ to evaluate model's explanatory capacity.

To evaluate robustness to data sparsity, we simulate missing chlorine concentration values by randomly masking chlorine concentration variable in the node features. Masking ratios vary from 0.0 to 1.0 in increments of 0.1. The same set of nodes is masked across all graphs to maintain spatial consistency for each ratio. Masked values are treated as missing during training. This setup reflects real-world partial observability while allowing for controlled comparison across masking levels.

We evaluate the model's computational cost under two threshold masking ratios 0 and 1, and expect the cost for other masking scenarios to fall within this range. Our AttentionR-GCN model was trained with a NVIDIA L40S GPU. At a masking ratio of 0, training used 45.65 MB memory, completed in 665 seconds with all 30 epochs completed, and achieved an average inference time of 1.70 seconds per batch. At a masking ratio of 1, training used 45.4 MB memory, completed in 354 seconds with early stop at 16 epochs. It has an average inference time of 0.05 seconds per batch.

## 5 RESULTS AND DISCUSSION

### 5.1 PERFORMANCE OF BASELINE MODELS WITHOUT TRANSFORMER ON ALL NODES

We compare attentionR-GCN with two ablations: (i) without the Transformer and (ii) without both the Transformer and the learnable embedding, against standard baselines. Across all three evaluation metrics, attentionR-GCN consistently outperforms the baselines Fig A1. At low mask ratios, GATv2, R-GCN, and r-GAT perform similarly. However, starting at mask ratio 0.5, attentionR-GCN outperform these methods, and the performance gap widens as data sparsity increases. The addition of the learnable embedding provides only minimal improvement, indicating that the main enhancements come from the relation-aware attention mechanism. These findings suggest that relation-based attention utilizes available features more effectively under high data sparsity than other approaches. Additionally, under sparse supervision, we consistently observe RGAT performing worse than simple relational baselines. This supports the original study Busbridge et al. (2019)'s conclusion that RGAT is highly task-dependent and struggles to learn stable attention when input signals are weak (few or low-quality features, limited labels). In such cases, R-GCN and related spectral baselines tend to generalize better, probably because they avoid the extra estimation required for attention parameters and thus overfit less when data is limited.

### 5.2 PERFORMANCE OF TRANSFORMER-ENHANCED, EDGE-AWARE MODELS ON ALL NODES

Fig. A1 shows that models including edge information generally outperform the edge-agnostic models, except for RGAT. Therefore, we add a Transformer module to the edge-aware baselines and compare them with our full model, which includes both the learnable embedding and the Transformer. As shown in Fig. 2, our model continues to outperform these models. However, TransformerR-GCN, TransformerGATv2, and TransformerrGAT can sometimes achieve similar performance to our model. Specifically, for MAE, they consistently perform similarly to our model. For MSE and $R^2$, they show comparable performance to our model at very low masking ratios (0–0.2) and very high masking ratios (0.8–1.0). However, in the middle range (0.2–0.8), our model consistently performs better. This pattern aligns with an information-availability perspective: when observations are plentiful, all methods perform well; under extreme sparsity, all are similarly limited by scarce information; and in the intermediate range, differences in how models exploit partial observations and heterogeneous relations become decisive. Therefore, the consistent better performance of our model in the medium masking ratio demonstrates that the relation-aware attention more effectively leverages available heterogeneous edge and node information. Additionally, similar MAE but lower MSE shows that our model better reduces large errors while keeping comparable typical absolute errors as the other three models. Higher $R^2$ indicates that our model explains a larger portion of target variability under partial observability than the three baselines.

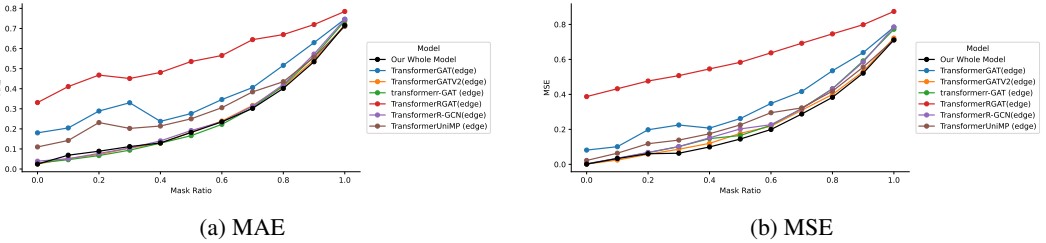

(a) MAE           (b) MSE

Figure 2: Test-set performance of Transformer variants across masking ratios

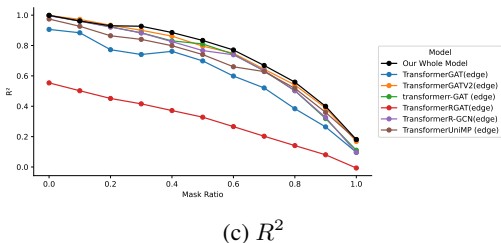

(c) $R^2$

Figure 2: Test-set performance of Transformer variants across masking ratios (continued)

## 5.3 PERFORMANCE ON MASKED AND UNMASKED NODES

Figure 3 and Figure A2 shows that all models perform significantly better on unmasked nodes than on masked nodes, which is expected given the greater supervision available for unmasked nodes during training. Our model outperforms all transformer baselines on both masked and unmasked nodes. However, for the unmasked subset, TransformerR-GCN, TransformerGATv2, and TransformerrGAT achieve performance comparable to ours, with very low MAE and MSE and high $R^2$ across all masking ratios. For masked nodes, however, our model consistently outperforms these models, with the largest margins in the intermediate masking range (0.2–0.8). This pattern supports earlier findings, indicating that the overall performance improvement of our model over the other three is mainly due to better results on masked nodes rather than unmasked ones. Such results suggest that our relation-aware attention design provides stronger inductive generalization to unobserved nodes, enabling more reliable interpolation. This is important for inferring unknown nodes in the networks under partial observability.

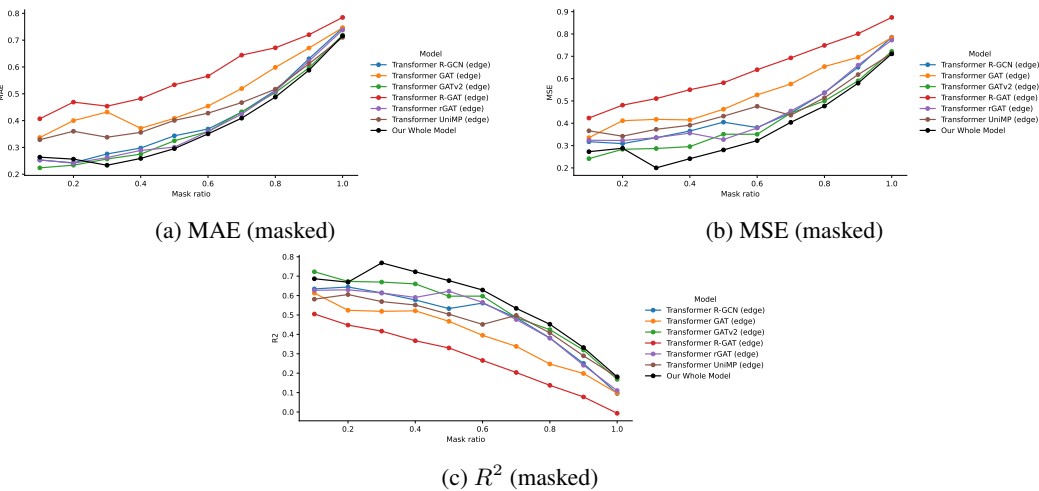

(a) MAE (masked)             (b) MSE (masked)

(c) $R^2$ (masked)

Figure 3: Performance on masked nodes across three metrics (MAE, MSE, $R^2$).

## 5.4 ABLATION TEST

Figure A3 shows an ablation study of the proposed model, starting with the naive attentionR-GCN and incrementally adding components, evaluated across different masking ratios and metrics. Across all masking levels, all variants demonstrate similar MAE. Adding the Transformer module produces modest yet consistent improvements in MSE and $R$ within the medium masking range (0.2–0.8). These results suggest that the attentionR-GCN backbone provides the main performance improvements, while the Transformer offers additional benefits under partial observability, especially at intermediate masking levels.

## 5.5 UNCERTAINTY QUANTIFICATION AT INDIVIDUAL LEVELS

We compute the per-sample MAE and MSE on the test set. Figure 4 shows that, at low masking ratios, errors remain consistently small across samples. As the masking ratio increases, error magnitudes rise slightly but stay centered around small values. Only at very high masking ratios, heavier tails appear and the distribution becomes more uniform across both small and large errors. These patterns suggest model's reliable performance under low–to–moderate masking, with no significant outliers. Figure A4 shows the mean and standard deviation of MAE and MSE for different y values across various masking ratios. At low masking levels, errors are both small and consistent across the range of target values y. For medium to high masking, the model performs best at mid-range y values, while both error magnitude and variability increase at the low and high y values. This indicates that the model is most accurate within the central target range and less reliable near the edges for medium to high masking ratios.

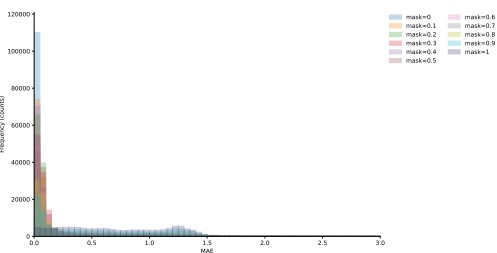 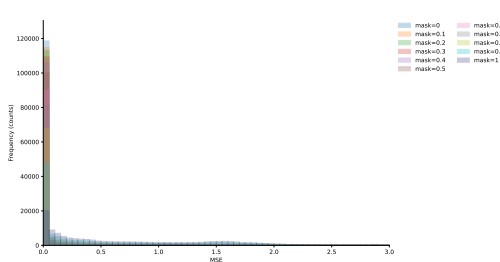

(a) Distribution of MAE across samples for test set under different masking ratios

(b) Distribution of MSE across samples for test set under different masking ratios

Figure 4: Error distribution over individual samples for the test set under different masking ratios

## 5.6 UNCERTAINTY QUANTIFICATION OVER DIFFERENT SEEDS

Figure A5 presents the mean MAE and MSE with standard deviations across three random seeds. The small standard deviations demonstrate robust, seed-stable performance. Figure A6 further distinguishes results between masked and unmasked nodes. While variability remains low overall, masked nodes show higher mean errors and larger standard deviations than unmasked nodes, indicating greater sensitivity in the masked setting.

## 6 CONCLUSION

We introduce AttentionR-GCN, a model for accurate forecasting in partially observed, dynamically evolving infrastructure networks. AttentionR-GCN tackles core limitations of prior work by integrating: (i) relation-aware, attention-driven aggregation that leverages heterogeneous node and edge features; (ii) learnable embeddings for missing features; and (iii) a Transformer-based temporal encoder that captures long-range temporal dependencies. Empirical results on two simulated water-distribution networks show that AttentionR-GCN outperforms strong baselines, with the largest gains at moderate sparsity. The model generalizes well, maintaining robust accuracy on fully unmonitored nodes at medium masking ratios. Ablations reveal that most of the improvement comes from the relation-aware attention, with small incremental benefits from the learnable missing-feature embeddings and the Transformer module. Finally, uncertainty quantification and cross-seed validation validate the model's robuestness. Together, these findings highlight the value of adaptive weighting of relational edge and node features, explicit missing-data handling, and temporal modeling for robust spatiotemporal forecasting.

Although our results demonstrate the promise of AttentionR-GCN, our evaluation is limited to a water infrastructure network. Further studies on diverse infrastructure datasets, such as energy grids or transportation networks, would provide additional validation of its generalizability and robustness. Exploring extensions to real-world deployments and examining aspects of model interpretability represent valuable directions for future work.

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

# A APPENDIX

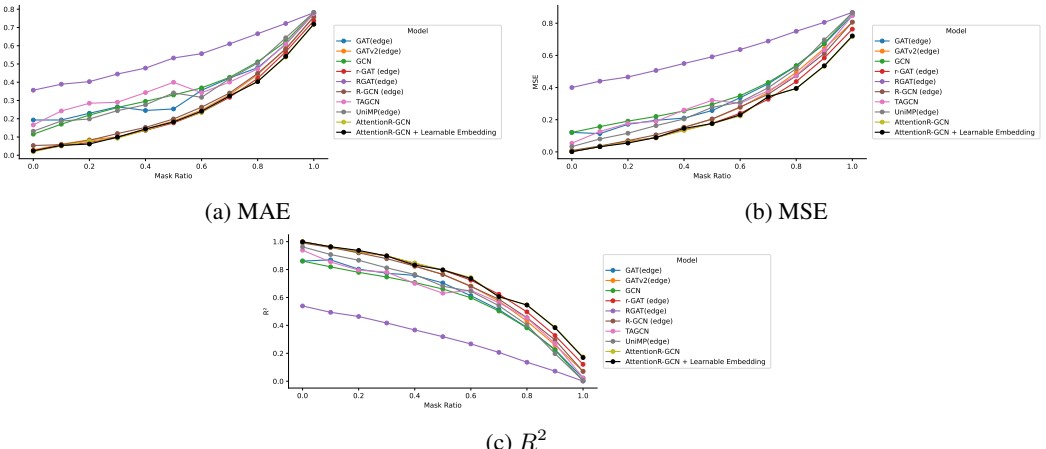

(a) MAE

(b) MSE

(c) $R^2$

Figure A1: Comparison of basic model performance across masking ratios on test set

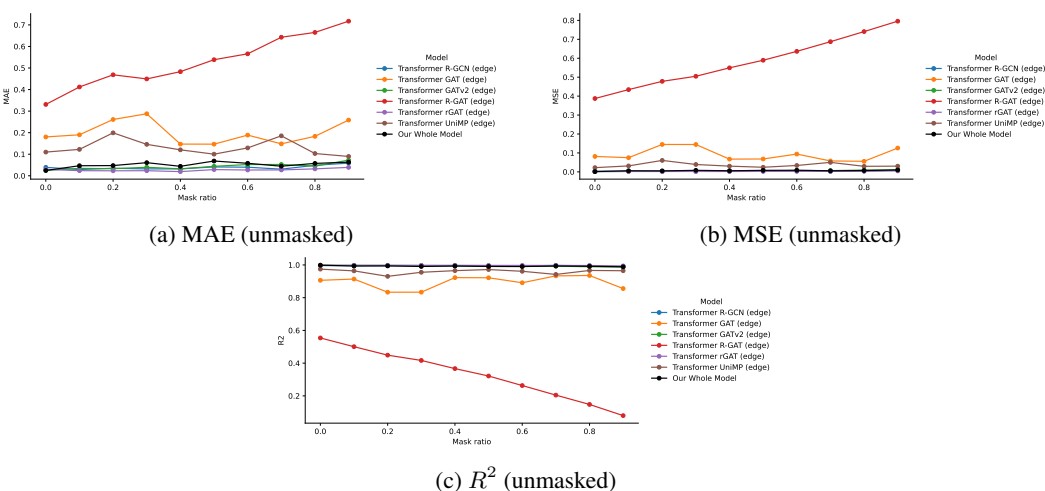

(a) MAE (unmasked)

(b) MSE (unmasked)

(c) $R^2$ (unmasked)

Figure A2: Performance on unmasked nodes across three metrics (MAE, MSE, $R^2$).

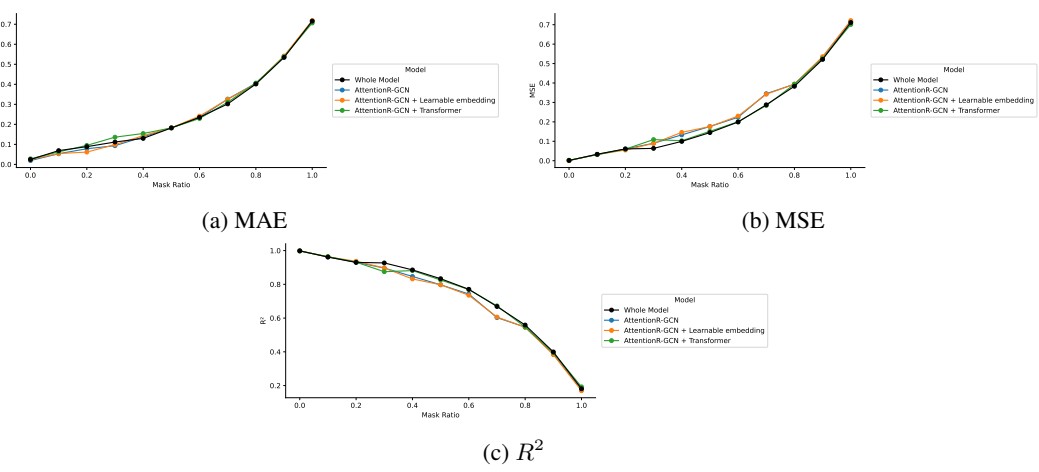

(a) MAE

(b) MSE

(c) $R^2$

Figure A3: Ablation test on three metrics

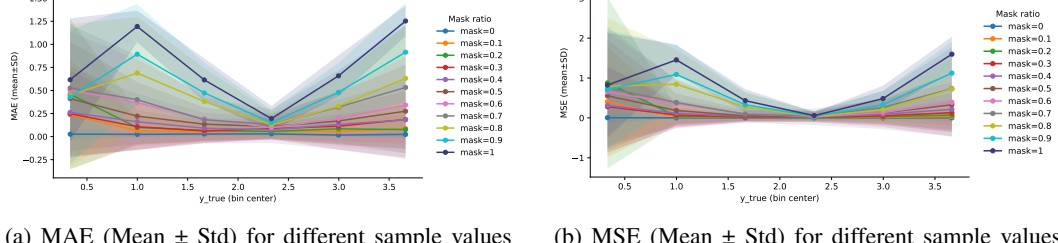

(a) MAE (Mean ± Std) for different sample values across mask ratios

(b) MSE (Mean ± Std) for different sample values across mask ratios

Figure A4: MAE and MSE with their mean and standard deviation for different sample values on the test sets across different mask ratio.

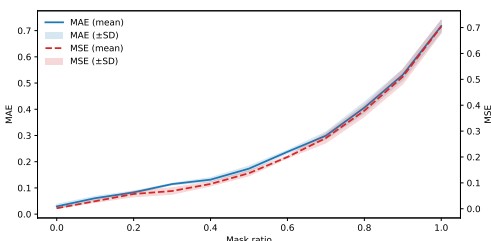

Figure A5: MAE and MSE on mean and std on all nodes on test sets over different seeds

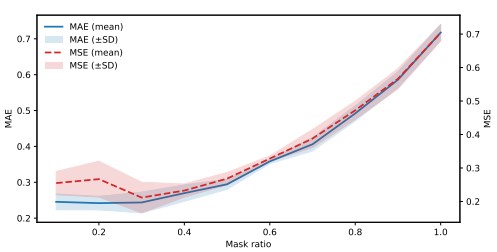

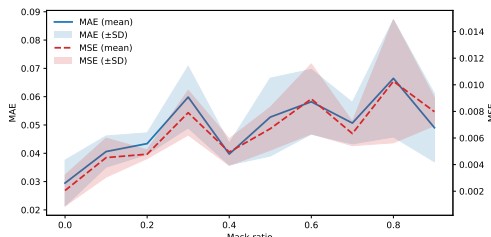

(a) MAE and MSE on mean and std masked nodes on test sets over different seeds

(b) MAE and MSE mean and std on unmasked nodes on test sets over different seeds

Figure A6: Evaluation of MAE and MSE across seeds, reported separately for masked and unmasked nodes; note the much smaller y-axis scales in the unmasked graph than the masked graph.

