# OpenReview forum: "AttentionR-GCN: Incorporating Spatiotemporal Reasoning in Heterogeneous and Partially Observed Graphs"
_ICLR.cc/2026/Conference — Submitted to ICLR 2026_

### Official Review · Reviewer_r4ka · 2025-10-25

**Soundness:** 2
**Presentation:** 1
**Contribution:** 1
**Rating:** 2
**Confidence:** 4

**Summary:**

This paper proposes AttentionR-GCN, for spatiotemporal prediction in partially observable, dynamically evolving urban infrastructure networks. The model combines a triple-based relation-aware attention mechanism, a learnable missing value embedding, and a Transformer temporal encoder to capture long-term temporal dependencies. Extensive experiments on two simulated water network datasets demonstrate the promising performance of AttentionR-GCN.

**Strengths:**

1. A unified relationship-aware attention mechanism is proposed to jointly model nodes, edges, and relationship types, which is an extension of the existing GAT and R-GCN.
2.   The model architecture and overall pipeline are relatively easy to understand.

**Weaknesses:**

1. Overall,  the paper suffers from poor presentation quality. Figures are unclear, tables summarizing experimental results are missing, and reference formatting is inconsistent. Moreover, this paper lacks the statement on the Use of LLMs.
2. The proposed components mainly combine existing techniques. The paper lacks a clear theoretical justification or new algorithmic insight that distinguishes it from prior work.
3.  Experiments are conducted only on two simulated datasets, without evaluation on real-world data. The lack of ablation studies, sensitivity analyses, and implementation details, e.g., hyperparameters, computational efficiency, prevents a full assessment of the model’s robustness and generalization.

**Questions:**

1. What is the computational complexity of AttentionR-GCN? Please provide the total number of trainable parameters, as well as training and inference runtimes. Such information is essential for assessing the scalability and reproducibility of the model.
2. The paper lacks comparisons with recent state-of-the-art edge-aware attention and temporal-GNN models.

---

### Official Review · Reviewer_3Sxo · 2025-10-30

**Soundness:** 2
**Presentation:** 2
**Contribution:** 2
**Rating:** 2
**Confidence:** 4

**Summary:**

This paper proposes an extension of the graph attention network based on different relation types. They use learnable embeddings to represent missing values and incorporate a transformer encoder to model temporal dependencies. They demonstrated its performance using public benchmarks and compared it with other baseline methods.

**Strengths:**

The authors focused on the multiple relational characteristics, sparsity, and time dependence of urban spatiotemporal data and constructed a relation-aware version of GATv2.

**Weaknesses:**

1.	The author only provided line graphs of the results of the proposed method on various data sets, but did not provide tabular statistical indicator results, which reduced the credibility of the paper.
2.	The authors did not provide a description of the use of LLM.
3.	The author's innovation is incremental and is a simple extension of GATv2.
4.	All the formulas in the paper are not numbered and there are many typos.

**Questions:**

1.	Should the comma in $S^{,r}_{u,v}[he]$ be deleted?
2.	Should there be a section describing the notion and problem definition you used?
3.	Should the results of the ablation experiment be given in a clearer table with specific quantitative values?
4.	Should the specific performance of the problem under different missing rates be displayed in the form of a table?
5.	Should some case studies be given to analyze the various relations learned?

---

### Official Review · Reviewer_h5m9 · 2025-11-01

**Soundness:** 3
**Presentation:** 2
**Contribution:** 2
**Rating:** 4
**Confidence:** 5

**Summary:**

The paper introduces AttentionR-GCN, a spatio-temporal GNN for heterogeneous, partially observed infrastructure graphs. The model employs relation-aware attention that jointly conditions on the self node, neighbor node, edge attributes, and relation type, utilizes learnable embeddings to handle missing values, and integrates the graph stack with a Transformer temporal encoder to capture long-range dynamics.

Experiments on simulated water distribution networks (C-Town and L-Town) evaluate one-step-ahead chlorine concentration forecasting under varying mask ratios, with results showing consistent gains over R-GCN, (r)GAT/(RGAT), UniMP, and Transformer-augmented counterparts, particularly at medium to high sparsity and on unmonitored nodes. Ablations suggest most of the gains stem from the relation-aware attention; uncertainty analyses over seeds and per-sample errors indicate stable behavior.

**Strengths:**

- Well-motivated problem (heterogeneous, partially observed urban networks), concise related-work positioning, and transparent training protocol (objectives, schedules, early stopping).

- Clear architectural specification (algorithmic form of logits/normalization, layer stack, figure) and controlled baselines with/without Transformer and edge features.

**Weaknesses:**

- Evaluation is confined to two simulated water networks with synthetic scenarios; no real-world telemetry or cross-domain tests (e.g., power grids, traffic) are provided. This limits claims of broad applicability.

- Focuses on one-step forecasting and general ML metrics (MAE/MSE/R2); there’s no multi-horizon evaluation or physics-aware constraints (e.g., mass-balance consistency, bounded chlorine kinetics), which matter in operations.

- The same set of nodes is masked across all graphs at a given ratio, which can induce distributional alignment and understate difficulty versus random per-instance masking or structured outages (blocks over space/time).

- While (r)GAT/R-GCN/UniMP and Transformer variants are included, influential spatio-temporal baselines like, DCRNN, STGCN, or Graph WaveNet with edge-aware adaptations aren’t reported head-to-head, despite being discussed in related work. A direct comparison would strengthen the empirical case. Other recent baselines such as GAP-LSTM are not even discussd.

**Questions:**

- How does performance change with random per-graph masking, temporally contiguous gaps, or spatially clustered outages (e.g., sensor bank down)? Results with these settings would better reflect real deployments.

- Have you tried multi-horizon forecasting? A short study could reveal whether the Transformer encoder retains advantages beyond one step.

- Could you incorporate simple physics-guided constraints or penalty terms (e.g., non-negativity, bounded concentration, smoothing

- Since attention is relation-aware, can you visualize per-relation attention maps and show case studies where specific relations or edge attributes dominate inference at masked nodes?

- Many edge attributes (flow/velocity) are noisy; what is the model performance under noised or biased edge features? A stress test would clarify fragility.

- See weaknesses.

---

### Meta-Review · Area_Chair_qyxc · 2026-01-07

**Summary:**

**Summary:**
This paper introduces AttentionR-GCN, a spatiotemporal gin that is targeted towards heterogeneous urban infrastructure networks. The model combines three previously characterized components, 1) a relation-aware attention mechanism, 2) learnable embeddings, and 3) a transformer encoder to capture temporal dependencies; and uses these to construct their method. On two simulated datasets, the authors show fairly significant improvements over baselines.

**Rationale:**
While the paper addresses an important and relevant problem, and the proposed method is reasonable, the work suffers from poor execution and poor presentation. The evaluations are limited and most relevant baselines are missing. The reviewers were universally negative on this work and I have to agree. I believe that the problem and work are interesting, and could be publishable if the reviewer feedback was taken into account earnestly, but not in its current form.

**Reviewer Concerns:**

Note, the authors did not engage with the reviewers so all concerns were dropped.

h5m9:
- narrow evaluation
- missing baselines

3Sxo:
- incremental novelty
- presentation issues

r4ka:
- poor novelty
- poor presentation quality
- missing computational analysis

**Reviewer Scores:**

h5m9, 4->4
3Sxo, 2->2
r4ka, 2->2

---

### Decision · Program_Chairs · 2026-01-26

Reject